# Unsupervised Learning for Physical Interaction through Video Prediction

**Chelsea Finn**[*]
UC Berkeley
cbfinn@eecs.berkeley.edu

**Ian Goodfellow**
OpenAI
ian@openai.com

**Sergey Levine**
Google Brain
UC Berkeley
slevine@google.com

## Abstract

A core challenge for an agent learning to interact with the world is to predict how its actions affect objects in its environment. Many existing methods for learning the dynamics of physical interactions require labeled object information. However, to scale real-world interaction learning to a variety of scenes and objects, acquiring labeled data becomes increasingly impractical. To learn about physical object motion without labels, we develop an action-conditioned video prediction model that explicitly models pixel motion, by predicting a distribution over pixel motion from previous frames. Because our model explicitly predicts motion, it is partially invariant to object appearance, enabling it to generalize to previously unseen objects. To explore video prediction for real-world interactive agents, we also introduce a dataset of 59,000 robot interactions involving pushing motions, including a test set with novel objects. In this dataset, accurate prediction of videos conditioned on the robot's future actions amounts to learning a "visual imagination" of different futures based on different courses of action. Our experiments show that our proposed method produces more accurate video predictions both quantitatively and qualitatively, when compared to prior methods.

## 1 Introduction

Object detection, tracking, and motion prediction are fundamental problems in computer vision, and predicting the effect of physical interactions is a critical challenge for learning agents acting in the world, such as robots, autonomous cars, and drones. Most existing techniques for learning to predict physics rely on large manually labeled datasets (e.g. [18]). However, if interactive agents can use unlabeled raw video data to learn about physical interaction, they can autonomously collect virtually unlimited experience through their own exploration. Learning a representation which can predict future video without labels has applications in action recognition and prediction and, when conditioned on the action of the agent, amounts to learning a predictive model that can then be used for planning and decision making.

However, learning to predict physical phenomena poses many challenges, since real-world physical interactions tend to be complex and stochastic, and learning from raw video requires handling the high dimensionality of image pixels and the partial observability of object motion from videos. Prior video prediction methods have typically considered short-range prediction [17], small image patches [22], or synthetic images [20]. Such models follow a paradigm of reconstructing future frames from the internal state of the model. In our approach, we propose a method which does not require the model to store the object and background appearance. Such appearance information is directly available in the previous frame. We develop a predictive model which merges appearance information

---

[*]Work was done while the author was at Google Brain.

from previous frames with motion predicted by the model. As a result, the model is better able to predict future video sequences for multiple steps, even involving objects not seen at training time.

To merge appearance and predicted motion, we output the motion of pixels relative to the previous image. Applying this motion to the previous image forms the next frame. We present and evaluate three motion prediction modules. The first, which we refer to as dynamic neural advection (DNA), outputs a distribution over locations in the previous frame for each pixel in the new frame. The predicted pixel value is then computed as an expectation under this distribution. A variant on this approach, which we call convolutional dynamic neural advection (CDNA), outputs the parameters of multiple normalized convolution kernels to apply to the previous image to compute new pixel values. The last approach, which we call spatial transformer predictors (STP), outputs the parameters of multiple affine transformations to apply to the previous image, akin to the spatial transformer network previously proposed for supervised learning [11]. In the case of the latter two methods, each predicted transformation is meant to handle separate objects. To combine the predictions into a single image, the model also predicts a compositing mask over each of the transformations. DNA and CDNA are simpler and easier to implement than STP, and while all models achieve comparable performance, the object-centric CDNA and STP models also provide interpretable internal representations.

Our main contribution is a method for making long-range predictions in real-world videos by predicting pixel motion. When conditioned on the actions taken by an agent, the model can learn to imagine different futures from different actions. To learn about physical interaction from videos, we need a large dataset with complex object interactions. We collected a dataset of 59,000 robot pushing motions, consisting of 1.5 million frames and the corresponding actions at each time step. Our experiments using this new robotic pushing dataset, and using a human motion video dataset [10], show that models that explicitly transform pixels from previous frames better capture object motion and produce more accurate video predictions compared to prior state-of-the-art methods. The dataset, video results, and code are all available online: `sites.google.com/site/robotprediction`.

## 2   Related Work

**Video prediction:**   Prior work on video prediction has tackled synthetic videos and short-term prediction in real videos. Yuan et al. [30] used a nearest neighbor approach to construct predictions from similar videos in a dataset. Ranzato et al. proposed a baseline for video prediction inspired by language models [21]. LSTM models have been adapted for video prediction on patches [22], action-conditioned Atari frame predictions [20], and precipitation nowcasting [28]. Mathieu et al. proposed new loss functions for sharper frame predictions [17]. Prior methods generally reconstruct frames from the internal state of the model, and some predict the internal state directly, without producing images [23]. Our method instead transforms pixels from previous frames, explicitly modeling motion and, in the case of the CDNA and STP models, decomposing it over image segments. We found in our experiments that all three of our models produce substantially better predictions by advecting pixels from the previous frame and compositing them onto the new image, rather than constructing images from scratch. This approach differs from recent work on optic flow prediction [25], which predicts where pixels will move to using direct optical flow supervision. Boots et al. predict future images of a robot arm using nonparametric kernel-based methods [4]. In contrast to this work, our approach uses flexible parametric models, and effectively predicts interactions with objects, including objects not seen during training. To our knowledge, no previous video prediction method has been applied to predict real images with novel object interactions beyond two time steps into the future.

There have been a number of promising methods for frame prediction developed concurrently to this work [16]. Vondrick et al. [24] combine an adversarial objective with a multiscale, feedforward architecture, and use a foreground/background mask similar to the masking scheme proposed here. De Brabandere et al. [6] propose a method similar to our DNA model, but use a softmax for sharper flow distributions. The probabilistic model proposed by Xue et al. [29] predicts transformations applied to latent feature maps, rather than the image itself, but only demonstrates single frame prediction.

**Learning physics:**   Several works have explicitly addressed prediction of physical interactions, including predicting ball motion [5], block falling [2], the effects of forces [19, 18], future human interactions [9], and future car trajectories [26]. These methods require ground truth object pose information, segmentation masks, camera viewpoint, or image patch trackers. In the domain of reinforcement learning, model-based methods have been proposed that learn prediction on images [14, 27], but they have either used synthetic images or instance-level models, and have not demonstrated

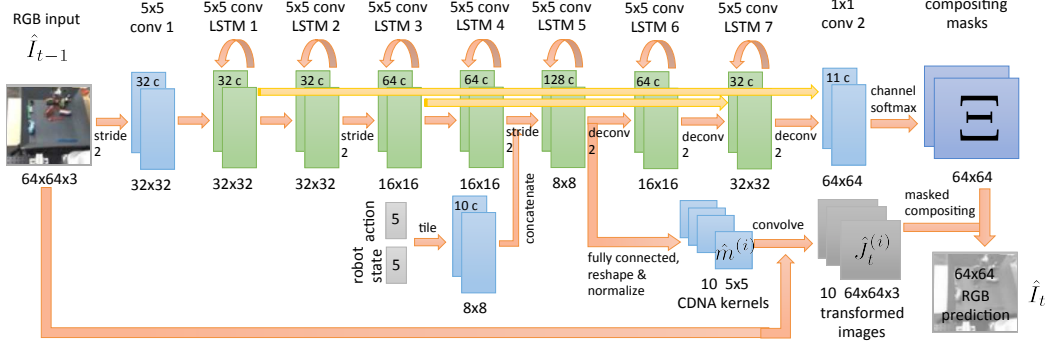

Figure 1: Architecture of the CDNA model, one of the three proposed pixel advection models. We use convolutional LSTMs to process the image, outputting 10 normalized transformation kernels from the smallest middle layer of the network and an 11-channel compositing mask from the last layer (including 1 channel for static background). The kernels are applied to transform the previous image into 10 different transformed images, which are then composited according to the masks. The masks sum to 1 at each pixel due to a channel-wise softmax. Yellow arrows denote skip connections.

generalization to novel objects nor accurate prediction on real-world videos. As shown by our comparison to LSTM-based prediction designed for Atari frames [20], models that work well on synthetic domains do not necessarily succeed on real images.

**Video datasets:** Existing video datasets capture YouTube clips [12], human motion [10], synthetic video game frames [20], and driving [8]. However, to investigate learning visual physics prediction, we need data that exhibits rich object motion, collisions, and interaction information. We propose a large new dataset consisting of real-world videos of robot-object interactions, including complex physical phenomena, realistic occlusions, and a clear use-case for interactive robot learning.

## 3 Motion-Focused Predictive Models

In order to learn about object motion while remaining invariant to appearance, we introduce a class of video prediction models that directly use appearance information from previous frames to construct pixel predictions. Our model computes the next frame by first predicting the motions of image segments, then merges these predictions via masking. In this section, we discuss our novel pixel transformation models, and propose how to effectively merge predicted motion of multiple segments into a single next image prediction. The architecture of the CDNA model is shown in Figure 1. Diagrams of the DNA and STP models are in Appendix B.

### 3.1 Pixel Transformations for Future Video Prediction

The core of our models is a motion prediction module that predicts objects' motion without attempting to reconstruct their appearance. This module is therefore partially invariant to appearance and can generalize effectively to previously unseen objects. We propose three motion prediction modules:

**Dynamic Neural Advection (DNA):** In this approach, we predict a distribution over locations in the previous frame for each pixel in the new frame. The predicted pixel value is computed as an expectation under this distribution. We constrain the pixel movement to a local region, under the regularizing assumption that pixels will not move large distances. This keeps the dimensionality of the prediction low. This approach is the most flexible of the proposed approaches.

Formally, we apply the predicted motion transformation $\hat{m}$ to the previous image prediction $\hat{I}_{t-1}$ for every pixel $(x, y)$ to form the next image prediction $\hat{I}_t$ as follows:

$$\hat{I}_t(x, y) = \sum_{k \in (-\kappa, \kappa)} \sum_{l \in (-\kappa, \kappa)} \hat{m}_{xy}(k, l) \hat{I}_{t-1}(x - k, y - l)$$

where $\kappa$ is the spatial extent of the predicted distribution. This can be implemented as a convolution with untied weights. The architecture of this model matches the CDNA model in Figure 1, except that

the higher-dimensional transformation parameters $\hat{m}$ are outputted by the last (conv 2) layer instead of the LSTM 5 layer used for the CDNA model.

**Convolutional Dynamic Neural Advection (CDNA):**   Under the assumption that the same mechanisms can be used to predict the motions of different objects in different regions of the image, we consider a more object-centric approach to predicting motion. Instead of predicting a different distribution for each pixel, this model predicts multiple discrete distributions that are each applied to the entire image via a convolution (with tied weights), which computes the expected value of the motion distribution for every pixel. The idea is that pixels on the same rigid object will move together, and therefore can share the same transformation. More formally, one predicted object transformation $\hat{m}$ applied to the previous image $I_{t-1}$ produces image $\hat{J}_t$ for each pixel $(x, y)$ as follows:

$$\hat{J}_t(x, y) = \sum_{k \in (-\kappa, \kappa)} \sum_{l \in (-\kappa, \kappa)} \hat{m}(k, l) \hat{I}_{t-1}(x - k, y - l)$$

where $\kappa$ is the spatial size of the normalized predicted convolution kernel $\hat{m}$. Multiple transformations $\{\hat{m}^{(i)}\}$ are applied to the previous image $\hat{I}_{t-1}$ to form multiple images $\{\hat{J}_t^{(i)}\}$. These output images are combined into a single prediction $\hat{I}_t$ as described in the next section and show in Figure 1.

**Spatial Transformer Predictors (STP):**   In this approach, the model produces multiple sets of parameters for 2D affine image transformations, and applies the transformations using a bilinear sampling kernel [11]. More formally, a set of affine parameters $\hat{M}$ produces a warping grid between previous image pixels $(x_{t-1}, y_{t-1})$ and generated image pixels $(x_t, y_t)$.

$$\begin{pmatrix} x_{t-1} \\ y_{t-1} \end{pmatrix} = \hat{M} \begin{pmatrix} x_t \\ y_t \\ 1 \end{pmatrix}$$

This grid can be applied with a bilinear kernel to form an image $\hat{J}_t$:

$$\hat{J}_t(x_t, y_t) = \sum_k^W \sum_l^H \hat{I}_{t-1}(k, l) \max(0, 1 - |x_{t-1} - k|) \max(0, 1 - |y_{t-1} - l|)$$

where $W$ and $H$ are the image width and height. While this type of operator has been applied previously only to supervised learning tasks, it is well-suited for video prediction. Multiple transformations $\{\hat{M}^{(i)}\}$ are applied to the previous image $\hat{I}_{t-1}$ to form multiple images $\{\hat{J}_t^{(i)}\}$, which are then composited based on the masks. The architecture matches the diagram in Figure 1, but instead of outputting CDNA kernels at the LSTM 5 layer, the model outputs the STP parameters $\{\hat{M}^{(i)}\}$.

All of these models can focus on learning physics rather than object appearance. Our experiments show that these models are better able to generalize to unseen objects compared to models that reconstruct the pixels directly or predict the difference from the previous frame.

### 3.2   Composing Object Motion Predictions

CDNA and STP produce multiple object motion predictions, which need to be combined into a single image. The composition of the predicted images $\hat{J}_t^{(i)}$ is modulated by a mask $\Xi$, which defines a weight on each prediction, for each pixel. Thus, $\hat{I}_t = \sum_c \hat{J}_t^{(c)} \circ \Xi_c$ , where $c$ denotes the channel of the mask and the element-wise multiplication is over pixels. To obtain the mask, we apply a channel-wise softmax to the final convolutional layer in the model (conv 2 in Figure 1), which ensures that the channels of the mask sum to 1 for each pixel position.

In practice, our experiments show that the CDNA and STP models learn to mask out objects that are moving in consistent directions. The benefit of this approach is two-fold: first, predicted motion transformations are reused for multiple pixels in the image, and second, the model naturally extracts a more object centric representation in an unsupervised fashion, a desirable property for an agent learning to interact with objects. The DNA model lacks these two benefits, but instead is more flexible as it can produce independent motions for every pixel in the image.

For each model, including DNA, we also include a "background mask" where we allow the models to copy pixels directly from the previous frame. Besides improving performance, this also produces interpretable background masks that we visualize in Section 5. Additionally, to fill in previously occluded regions, which may not be well represented by nearby pixels, we allowed the models to generate pixels from an image, and included it in the final masking step.

### 3.3 Action-conditioned Convolutional LSTMs

Most existing physics and video prediction models use feedforward architectures [17, 15] or feedforward encodings of the image [20]. To generate the motion predictions discussed above, we employ stacked convolutional LSTMs [28]. Recurrence through convolutions is a natural fit for multi-step video prediction because it takes advantage of the spatial invariance of image representations, as the laws of physics are mostly consistent across space. As a result, models with convolutional recurrence require significantly fewer parameters and use those parameters more efficiently.

The model architecture is displayed in Figure 1 and detailed in Appendix B. In an interactive setting, the agent's actions and internal state (such as the pose of the robot gripper) influence the next image. We integrate both into our model by spatially tiling the concatenated state and action vector across a feature map, and concatenating the result to the channels of the lowest-dimensional activation map. Note, though, that the agent's internal state (i.e. the robot gripper pose) is only input into the network at the beginning, and must be predicted from the actions in future timesteps. We trained the networks using an $l_2$ reconstruction loss. Alternative losses, such as those presented in [17] could complement this method.

## 4 Robotic Pushing Dataset

One key application of action-conditioned video prediction is to use the learned model for decision making in vision-based robotic control tasks. Unsupervised learning from video can enable agents to learn about the world on their own, without human involvement, a critical requirement for scaling up interactive learning. In order to investigate action-conditioned video prediction for robotic tasks, we need a dataset with real-world physical object interactions. We collected a new dataset using 10 robotic arms, shown in Figure 2, pushing hundreds of objects in bins, amounting to 57,000 interaction sequences with 1.5 million video frames. Two test sets, each with 1,250 recorded motions, were also collected. The first test set used two different subsets of the objects pushed during training. The second test set involved two subsets of objects, none of which

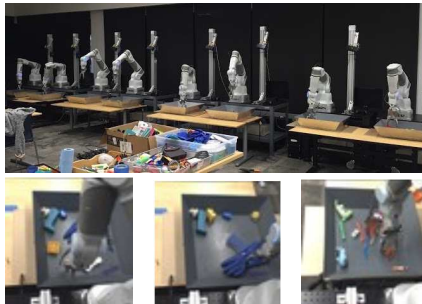

Figure 2: Robot data collection setup (top) and example images captured from the robot's camera (bottom).

were used during training. In addition to RGB images, we also record the corresponding gripper poses, which we refer to as the internal state, and actions, which corresponded to the commanded gripper pose. The dataset is publically available[2]. Further details on the data collection procedure are provided in Appendix A.

## 5 Experiments

We evaluate our method using the dataset in Section 4, as well as on videos of human motion in the Human3.6M dataset [10]. In both settings, we evaluate our three models described in Section 3, as well as prior models [17, 20]. For CDNA and STP, we used 10 transformers. While we show stills from the predicted videos in the figures, the qualitative results are easiest to compare when the predicted videos can be viewed side-by-side. For this reason, we encourage the reader to examine the video results on the supplemental website[2]. Code for training the model is also available on the website.

**Training details:** We trained all models using the TensorFlow library [1], optimizing to convergence using ADAM [13] with the suggested hyperparameters. We trained all recurrent models with and without scheduled sampling [3] and report the performance of the model with the best validation error. We found that scheduled sampling improved performance of our models, but did not substantially affect the performance of ablation and baseline models that did not model pixel motion.

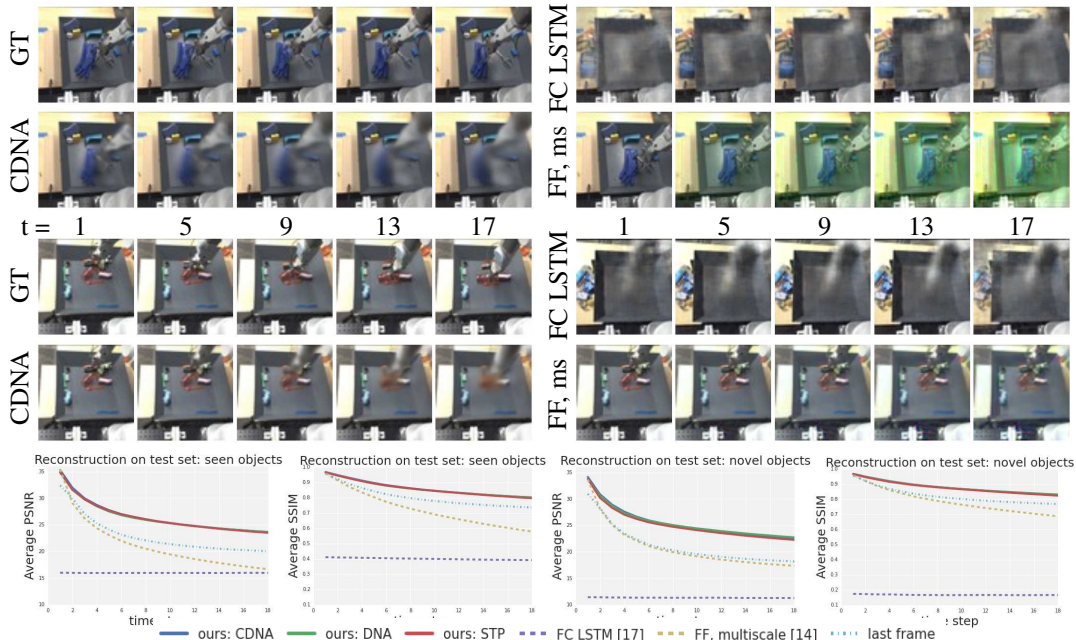

Figure 3: Qualitative and quantitative reconstruction performance of our models, compared with [20, 17]. All models were trained for 8-step prediction, except [17], trained for 1-step prediction.

## 5.1 Action-conditioned prediction for robotic pushing

Our primary evaluation is on video prediction using our robotic interaction dataset, conditioned on the future actions taken by the robot. In this setting, we pass in two initial images, as well as the initial robot arm state and actions, and then sequentially roll out the model, passing in the future actions and the model's image and state prediction from the previous time step. We trained for 8 future time steps for all recurrent models, and test for up to 18 time steps. We held out 5% of the training set for validation. To quantitatively evaluate the predictions, we measure average PSNR and SSIM, as proposed in [17]. Unlike [17], we measure these metrics on the entire image. We evaluate on two test sets described in Section 4, one with objects seen at training time, and one with previously unseen objects.

Figure 3 illustrates the performance of our models compared to prior methods. We report the performance of the feedforward multiscale model of [17] using an $l_1$+GDL loss, which was the best performing model in our experiments – full results of the multi-scale models are in Appendix C. Our methods significantly outperform prior video prediction methods on all metrics. The FC LSTM model [20] reconstructs the background and lacks the representational power to reconstruct the objects in the bin. The feedforward multiscale model performs well on 1-step prediction, but performance quickly drops over time, as it is only trained for 1-step prediction. It is worth noting that our models are significantly more parameter efficient: despite being recurrent, they contain 12.5 million parameters, which is slightly less than the feedforward model with 12.6 million parameters and significantly less than the FC LSTM model which has 78 million parameters. We found that none of the models suffered from significant overfitting on this dataset. We also report the baseline performance of simply copying the last observed ground truth frame.

In Figure 4, we compare to models with the same stacked convolutional LSTM architecture, but that predict raw pixel values or the difference between previous and current frames. By explicitly modeling pixel motion, our method outperforms these ablations. Note that the model without skip connections is most representative of the model by Xingjian et al. [28]. We show a second ablation in Figure 5, illustrating the benefit of training for longer horizons and from conditioning on the action of the robot. Lastly, we show qualitative results in Figure 6 of changing the action of the arm to examine the model's predictions about possible futures.

For all of the models, the prediction quality degrades over time, as uncertainty increases further into the future. We use a mean-squared error objective, which optimizes for the mean pixel values. The

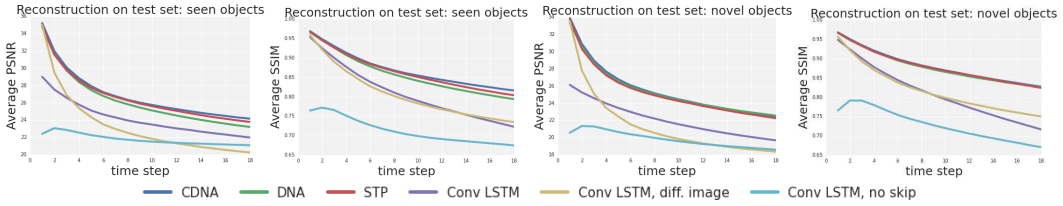

Figure 4: Quantitative comparison to models which reconstruct rather than predict motion. Notice that on the novel objects test set, there is a larger gap between models which predict motion and those which reconstruct appearance.

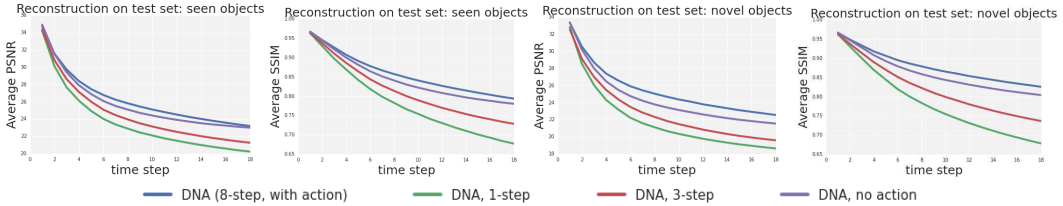

Figure 5: Ablation of DNA involving not including the action, and different prediction horizons during training.

model thus encodes uncertainty as blur. Modeling this uncertainty directly through, for example, stochastic neural networks is an interesting direction for future work. Note that prior video prediction methods have largely focused on single-frame prediction, and most have not demonstrated prediction of multiple real-world RGB video frames in sequence. Action-conditioned multi-frame prediction is a crucial ingredient in model-based planning, where the robot could mentally test the outcomes of various actions before picking the best one for a given task.

## 5.2 Human motion prediction

In addition to the action-conditioned prediction, we also evaluate our model on predicting future video without actions. We chose the Human3.6M dataset, which consists of human actors performing various actions in a room. We trained all models on 5 of the human subjects, held out one subject for validation, and held out a different subject for the evaluations presented here. Thus, the models have never seen this particular human subject or any subject wearing the same clothes. We subsampled the video down to 10 fps such that there was noticeable motion in the videos within reasonable time frames. Since the model is no longer conditioned on actions, we fed in 10 video frames and trained the network to produce the next 10 frames, corresponding to 1 second each. Our evaluation measures performance up to 20 timesteps into the future.

The results in Figure 7 show that our motion-predictive models quantitatively outperform prior methods, and qualitatively produce plausible motions for at least 10 timesteps, and start to degrade thereafter. We also show the masks predicted internally by the model for masking out the previous

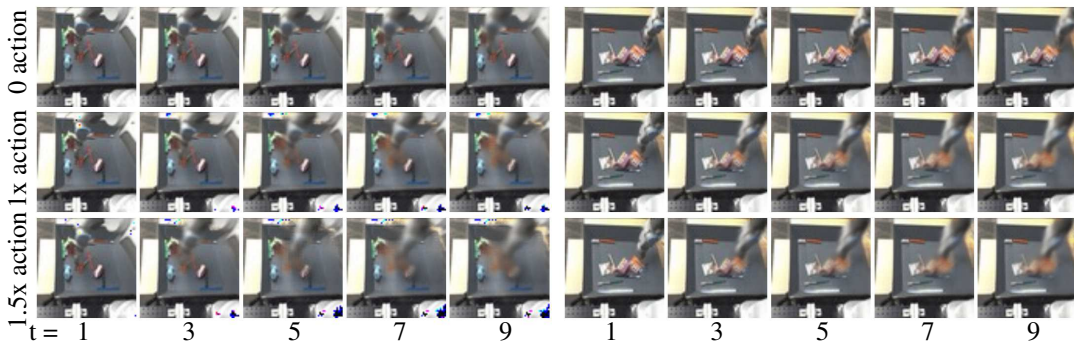

Figure 6: CDNA predictions from the same starting image, but different future actions, with objects *not seen in the training set*. By row, the images show predicted future with zero action (stationary), the original action, and an action 150% larger than the original. Note how the prediction shows no motion with zero action, and with a larger action, predicts more motion, including object motion.

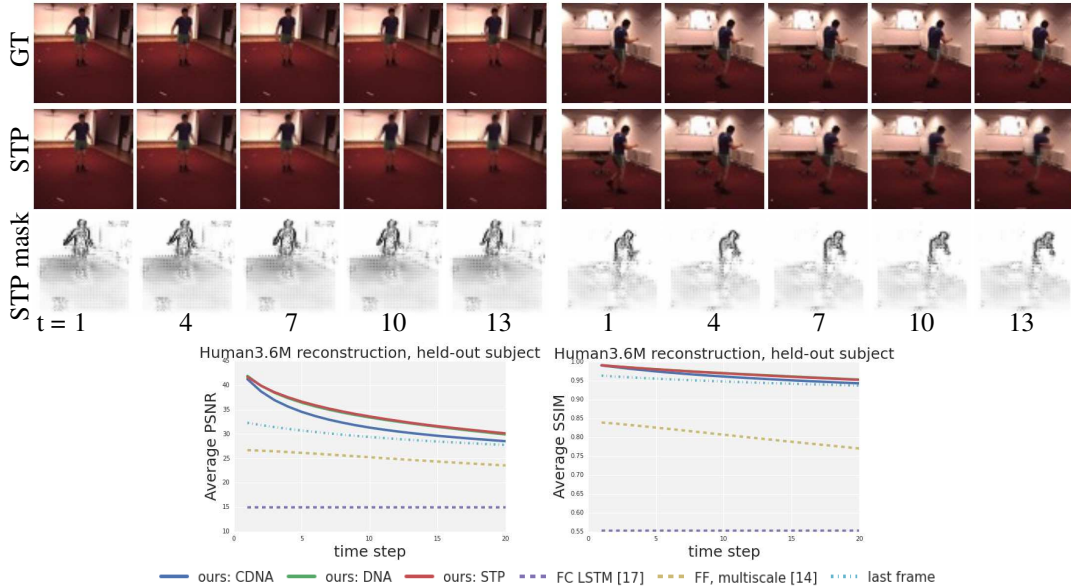

Figure 7: Quantitative and qualitative results on human motion video predictions with a *held-out human subject*. All recurrent models were trained for 10 future timesteps.

frame, which we refer to as the background mask. These masks illustrate that the model learns to segment the human subject in the image without any explicit supervision.

# 6    Conclusion & Future Directions

In this work, we develop an action-conditioned video prediction model for interaction that incorporates appearance information in previous frames with motion predicted by the model. To study unsupervised learning for interaction, we also present a new video dataset with $59,000$ real robot interactions and $1.5$ million video frames. Our experiments show that, by learning to transform pixels in the initial frame, our model can produce plausible video sequences more than 10 time steps into the future, which corresponds to about one second. In comparisons to prior methods, our method achieves the best results on a number of previous proposed metrics.

Predicting future object motion in the context of a physical interaction is a key building block of an intelligent interactive system. The kind of action-conditioned prediction of future video frames that we demonstrate can allow an interactive agent, such as a robot, to imagine different futures based on the available actions. Such a mechanism can be used to plan for actions to accomplish a particular goal, anticipate possible future problems (e.g. in the context of an autonomous vehicle), and recognize interesting new phenomena in the context of exploration. While our model directly predicts the motion of image pixels and naturally groups together pixels that belong to the same object and move together, it does not explicitly extract an internal object-centric representation (e.g. as in [7]). Learning such a representation would be a promising future direction, particularly for applying efficient reinforcement learning algorithms that might benefit from concise state representations.

### Acknowledgments

We would like to thank Vincent Vanhoucke, Mrinal Kalakrishnan, Jon Barron, Deirdre Quillen, and our anonymous reviewers for helpful feedback and discussions. We would also like to thank Peter Pastor for technical support with the robots.

## Footnotes

[2]See http://sites.google.com/site/robotprediction

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
