[Supplementary Material · nips_final_appendix_small.pdf]

# Appendix

## A  Data Collection Details

In this section, we discuss additional details of the data collection procedure.

The data was collected using 10 7-degree-of-freedom robot arms, and included the robot's joint angles, gripper pose, commanded gripper pose, measured torques, and $640 \times 512$ RGB images captured from the robot's camera. The images were center-cropped and downsampled to $64 \times 64$ for our experiments. We used smaller images for faster training. In principle, the proposed methods should be able to handle larger images if desired. The images and robot sensor readings were recorded at 10 Hz, and the robot was commanded via impedance control in task space. In our experiments, the robot's internal state was the gripper pose, and the action was the commanded gripper pose. Two test sets, each with $1{,}500$ recorded motions, were also collected. Both used the same robot control method described above. The first test set used two different subsets of the objects pushed during training. The second test set involved two subsets of objects, none of which were used during training.

During data collection, between ten and twenty random objects were placed in bins in front of each robot, and swapped out for new objects after approximately $4{,}000$ randomized interactions. The robots were programmed to repeatedly perform one of two different types of pushing motions: a random push or a randomized sweep to the middle. The sweep to the middle starts from a random position on the outside border of the bin, and meandered randomly towards the middle. The sweep motion was designed to prevent objects from piling up on the edges of the bin. Each type of motion lasted for approximately 3-5 seconds. Between each motion, the arm was programmed to move out of the camera scene, and an image was recorded.

## B  Model Details

In this section, we present additional details for each model. A diagram of the CDNA model is shown in the main paper in Figure 1, and additional diagrams for the DNA and STP models are shown in Figures 8 and 9, respectively. Each model consists of a core trunk made up of one stride-2 $5 \times 5$ convolution, followed by convolutional LSTMs. Each of these LSTM layers has the weights arranged into $5 \times 5$ convolutions, and the output of the preceding LSTM is fed directly into the next one. LSTM layers 3 and 5 are preceded by stride 2 downsampling to reduce resolution, and LSTM layers 5, 6, and 7 are preceded by $2\times$ upsampling. The end of the LSTM stack is followed by a $2\times$ upsampling stage and a final convolutional layer, which then outputs a full-resolution mask for compositing the various transformed predictions (in the case of the CDNA and STP) and compositing against the static background (in the case of all models, including the DNA). To preserve high-resolution

Figure 8: Architecture of the DNA model. In contrast to the other models, the DNA model outputs the spatially-varying transformation kernels from the last layer, rather than the middle convolutional LSTM layer. The kernels are applied to transform the previous image, and the transformed image is composited with a 2-channel foreground-background mask.

Figure 9: Architecture of the STP model. This model is identical to the CDNA, with the only difference being that instead of outputting 10 transformation kernels, the model outputs 10 affine transformation matrices (with an additional fully connected layer). As with the CDNA, the transformations are each applied to the previous image, and the 10 resulting transformed images are composited by using a mask.

information, skip connections are included from LSTM 1 to conv 2 and from LSTM 3 to LSTM 7. The skip connections simply concatenate the skip layer activations and those of the preceding layer before sending them to the following layer (e.g. the input to LSTM 7 consists of the concatenation of LSTM 6 and LSTM 3).

In the case of the action-conditioned robot manipulation task, all three models also include as input the current state and action of the robot (corresponding to gripper pose and gripper motion command). This 10-dimensional vector is first tiled into a $8 \times 8$ response map with $10$ channels, and then concatenated, channel-wise, to the input of LSTM 5. The next state is predicted linearly from the current state and action, though more sophisticated prediction models could be used for more complex systems.

The three models differ in the form of the transformation that is applied to the previous image. The object-centric CDNA and STP models output the transformation parameters after LSTM 5. In both cases, the output of LSTM 5 is flattened and linearly transformed, either directly into filter parameters in the case of the CDNA, or through one 100-unit hidden layer in the case of the STP. There are $10$ CDNA filters, which are $5 \times 5$ in size and normalized to sum to 1 via a spatial softmax, so that each filter represents a distribution over positions in the previous image from which a new pixel value can be obtained. The STP parameters correspond to $10 \; 3 \times 2$ affine transformation matrices. The transformations are applied to the preceding image to create 10 separate transformed images. The CDNA transformation corresponds to a convolution (though with the kernel being an output of the network), while the STP transformation is an affine transformation.

The DNA model differs from the other two in that the transformation parameters are outputted at the last layer, in the same place as the mask. This is because the DNA model outputs a transformation map as large as the entire image. For each image pixel, the model outputs a $5 \times 5$ convolutional kernel that can be applied to the previous image to obtain a new pixel value, similarly to the CDNA model. However, because the kernel is spatially-varying, this model is not equivalent to the CDNA. This transformation only produces one transformed image.

After transformation, the transformed image(s) and the previous image are composited together based on the mask. The previous image is included as a static "background" image and, as shown in Section 5, the mask on the background image indeed tends to pick out static parts of the scene. The final image is formed by multiplying each transformed image and the background image by their mask values, and adding all of the masked images together.

## C  Additional Experimental Results

Here we show additional experimental results.

Figure 10: Comparison of various losses using the feedforward multi-scale architecture from [17]. We were unable to get the adversarial objective to train desirably.

To evaluate the method of [17], we trained the feedforward multiscale architecture using four of the proposed objectives. For the robot motion dataset, we added the actions to the architecture at each scale via tiling. We were unable to successfully train the model with an adversarial loss, and, as shown in Figure 10, the model trained GDL+$l_1$ performed the best on the robot dataset. Thus, we presented the results of this model in the main evaluation.