[Reviews · NeurIPS 2016]

Reviewer 1

Summary

This paper develops novel deep network architectures that make pixel-level predictions of video frames, and introduces a new large scale dataset containing video of real-world object manipulation by robotic arms. The architectures all predict the motion of pixels in each frame, rather than directly predicting pixel values. This approach enables more accurate predictions on unseen objects. Two of the architectures also contain a ‘compositing’ scheme, whereby different transformations are applied to different objects, resulting in more or less unsupervised segmentation of objects. The large scale dataset enables end to end training of the system, and rigorous evaluation of competing approaches. Moreover, beyond the video input, the dataset contains the pose of the robot arm as well as a representation of the goal pose, making it suitable for learning about the effects of actions in the environment. Together these advances may help develop robots which can learn from unstructured interaction with their environment.

Qualitative Assessment

This paper introduces a very intriguing dataset of real world physical interactions between a robot and a variety of objects. This is a strong contribution which is likely to generate significant future work. The models proposed in the paper outperform competitor methods, though it should be noted that the absolute quality of predictions is still rather poor. The paper understandably focuses on making use of the real-world dataset it introduces. However to justify some of its claims that the neural network architectures it proposes are capable of learning the physics of the environment, it would be very helpful to test these architectures on, eg, the block world dataset of Battaglia et al. This would enable controlled examination of the degree to which the models learn genuine physical concepts and constraints. Said another way, learning physics (i.e. underlying causal structure) may be different from learning predictions of motion. There is very recent work which has made pixel-level predictions multiple time steps into the future, with results that seem at least comparable to what is achieved in this paper: Lotter, Kreiman, & Cox. Deep Predictive Coding Networks for Video Prediction and Unsupervised Learning. ArXiv, 2016. Ideally this method would be part of the quantitative comparisons in the paper, but given how recent this work is, that may not be possible. It should at a minimum be discussed, though.

Confidence in this Review

2-Confident (read it all; understood it all reasonably well)


Reviewer 2

Summary

The paper presents three deep learning methods for predicting future images of a scene. The authors learn a distribution of the pixels in the previous scene for each pixel. The first method predicts pixel movements and the other two are designed to implicitly estimate object movements. The network includes multiple masks including one for the background. The methods were evaluated on pushing data collected using multiple robots and the Human3.6M dataset.

Qualitative Assessment

The paper mentions appendices and a link to videos, but the link does not work and the appendices are not included in the main file nor the supplementary material. I am reviewing the paper as it is. The descriptions of the three methods need to be expanded. The method descriptions rely heavily on Fig. 1 and assume that most of the details are clear from the network structure. I did not find this to be the case. What is the structure of the m/M? Do the they adapt to the local image patches or are they constant over the entire image and rely on the masks? The authors should consider including a visualization of an m/M. Can they be interpreted? The amount of data collected using the robot setup is impressive. The methods also outperform the state of the art. That being said, the paper is missing a discussion of the methods’ weaknesses and limitations. For example, why is the image resolution 64x64 pixels? Is it possible to use higher resolutions? The pushing task mainly involves planar movements parallel to the image plane. Can the method be applied to more 3D movements like a can tipping over? The authors should consider showing an example of a larger turning motion from the Human3.6M dataset for figure 7. The qualitative results also leave room for improvement. There is a considerable amount of blurring in the images. The task of predicting videos is obviously very challenging and I would not expect perfect results. However, the authors need to discuss the sources of these errors and how they may be addressed in the future. The authors could consider exploiting the robot setup more for the evaluations. Can the robot grasp a moved object using only the predicted image? Is the quality of the prediction sufficient? The MSE criterion is based on the optical flow computed for the ground truth data and the predicted data. It would be helpful to include an example image showing the optical flow computed for both images. How well does the optical flow handle the blurring? Are there situations where the authors would recommend one of their proposed methods over the others? The authors could merge some of the plots to make additional space for discussions. How does the performance change when the background masks are removed? Overall, the paper presents an interesting method for a very challenging task. The authors also present a large new dataset for video prediction. The network structure is impressive and seems well thought through. I therefore think that the authors have done great work. This work is unfortunately not reflected in the paper. Adding more detailed descriptions and thorough discussions of the methods’ limitations would make the paper more accessible and significantly increase the paper’s long term impact.

Confidence in this Review

2-Confident (read it all; understood it all reasonably well)


Reviewer 3

Summary

The paper proposes a framework for making long-range predictions in videos by predicting motion. This is different from the standard approaches that reconstruct the next frame(s) explicitly from some internal state of the model. A weakness of such approaches is that the internal state needs to explicitly handel appearance information. However, this is actually not necessary, as this kind of information is readily available in the current input frame. The proposed method instead can focus learning the necessary physical concepts that concern motion. The motion prediction is implemented by a deep network that outputs either convolutional filters or affine transformation matrices that are applied to the current input frame. Additionally, the method allows to integrate conditional input in the form of actions/states of a robot interacting with the provided scenery.

Qualitative Assessment

Overall comment: The paper is well motivated and well written, nearly every presented aspect is understandable. (I do not completely understand how this 'tiling' operation of the action/input state works in detail). It might have been conceptually better to separate out the fact that the model is itself action-conditioned. While the motivation for introducing the model has a clear 'interaction' aspect, focusing only on predicting motion first would have made the two contributing aspects (prediction, action-conditioning) better visible (well, a personal opinion). Of course, without the action-part, the presented model has some similarities to the recently (NIPS-submit) Dynamic Filter Networks from Luc van Gools group. Another very recent suitable reference/related work should be Forecasting from Static Images using Variational Autoencoders out of CMU. I would have loved a more detailed analysis of things that actually do not work too well yet. E.g. it seems that mostly predicting the robot arm movement works, though the moved objects are still quite blurry. This seems to be particularly true for non-rigid objects (if my interpretation of the videos is correct)? This might be due to the fact that the training loss still is the reconstruction loss in pixel space? A more GAN-type loss could have a positive effect here? A very different question is related to the overall approach of producing predictions in the actual pixel space. Apart from the fact that the results are interpretable to humans, there does not seem to be a reason for that. Instead, predicting some (compact?) code for the next frames should be much more reasonable, e.g. with respect to the available capacaty of the model or with respect to the subsequent use of this predictive framework?

Confidence in this Review

2-Confident (read it all; understood it all reasonably well)


Reviewer 4

Summary

This paper presents three different approaches to predict future video frames conditional of an agent action in an unsupervised way. More precisely, the approach predicts how objects move in a video without explicitly learning the optical flow. The method relies on a deep network combining LSTM, dynamic neural advection and spatial transformers. The method is extensively evaluated on a new dataset of robots manipulating objects (to be released soon according to the authors) and on the Human3.6SM dataset. The results are compared with the state of the art, showing excellent performance.

Qualitative Assessment

This paper is well written, the method well evaluated and the quantitative results convincing. Learning physical interaction from video is indeed a challenging topic, which could greatly impact autonomous systems and beyond. Unfortunately, it is difficult to me to really assess the strength of the approach, despite the excellent quantitative results: 1) First, the still images are not convincing. The object to predict becomes quickly blurry. A prediction of 1s is quite limited, and if the image becomes already blurry at that stage, I wonder what would be the utility of the approach. That said, I could not look at the videos as the link was not working. Maybe the results are more convincing in the videos. 2) The use of LSTM for prediction is not really new. How did the author came up with such an architecture. Would adding more LSTM layer improve the prediction length? What about using differentiable long-term memory (like in NTM)? 3) For the dynamic neural advection, it is not clear to me what is M exactly. Is is a scalar value for each pixel? How is the motion parametrized? 4) It would have been great to provide more mathematical details on each part of the architecture, as this could greatly help the reader in grasping what were the modeling assumptions and limitations but also help reproduce the results. Unfortunately, the link with videos and all the appendices were not available at the time of this review.

Confidence in this Review

2-Confident (read it all; understood it all reasonably well)


Reviewer 5

Summary

The paper is about learning to predict the outcomes from the physical interaction in an unsupervised manner. The main idea is to predict the motion of the pixel (as opposed to predicting the actual pixels values). Building upon convolutional LSTM, the authors present three predictive models to achieve this goal (DNA, CDNA, and STP). These methods are evaluated on a new large-scale physical interaction dataset as well as the Human3.6M datasets. Quantitative comparisons against [14,17] are presented.

Qualitative Assessment

Clarity of exposition: - The paper is well-written and easy to understand. - The discussion of the literature is comprehensive, clear and well-organized. - The paper only shows the CDNA architecture. The other two architectures are described in the texts. I wonder if it is also possible to visualize the other two architectures in Figure 1 (or in the supplementary material). - In the supplementary site, the authors show video results as well as the masks. Method: The presented method achieves excellent results compared with other state-of-the-art algorithms. The predicted frame, however, suffers from blur artifacts and missing details. For examples, - It seems to me that the masks do not have temporal coherence. I wonder why this is the case. Does the frame prediction in the next frame depends on the current frame? - The collection of motion transformation M can be viewed as the backward optical flow field. It will be interesting to impose priors on the term (e.g., penalizing large gradients of the flow fields) to encourage spatial smoothness. In sum, this is well-written paper with solid results. The presented methods provide a new way to predict future frames from physical interaction. I think the paper makes solid contributions to the community.

Confidence in this Review

2-Confident (read it all; understood it all reasonably well)


Reviewer 6

Summary

The authors propose a model that predicts pixels in video conditioned on the actions. The authors also plan to provide the dataset of 50k robot interactions as part of the contribution. They evaluate the model not only on the robot interaction dataset but also on human motion prediction.

Qualitative Assessment

While the model is clear and the dataset creation part is clear the figures showing the predictions are very hard and it would have been great if the authors provided supplementary material of videos.

Confidence in this Review

2-Confident (read it all; understood it all reasonably well)